# Riparian Land Cover, Water Temperature Variability, and Thermal Stress for Aquatic Species in Urban Streams

**Anne Timm** [1],*[ID]**, Valerie Ouellet** [2][ID] **and Melinda Daniels** [2][ID]

1   USDA Forest Service, Northern Research Station, Baltimore, MD 21228, USA
2   Stroud Water Research Center, Avondale, PA 19311, USA; valeria.ouellet@gmail.com (V.O.); mdaniels@stroud.org (M.D.)
*   Correspondence: anne.l.timm@usda.gov

**Abstract:** Thermal regime warming and increased variability can result in human developed watersheds due to runoff over impervious surfaces and influence of stormwater pipes. This study quantified relationships between tree canopy, impervious surface, and water temperature in stream sites with 4 to 62% impervious land cover in their "loggersheds" to predict water temperature metrics relevant to aquatic species thermal stress thresholds. This study identified significant ($\geq 0.7$, $p < 0.05$) negative correlations between water temperature and percent tree canopy in the 5 m riparian area and positive correlations between water temperature and total length of stormwater pipe in the loggershed. Mixed-effects models predicted that tree canopy cover in the 5 m riparian area would reduce water temperatures 0.01 to 6 °C and total length of stormwater pipes in the loggershed would increase water temperatures 0.01 to 2.6 °C. To our knowledge, this is the first time that the relationship between stormwater pipes and water temperature metrics has been explored to better understand thermal dynamics in urban watersheds. The results highlight important aspects of thermal habitat quality and water temperature variability for aquatic species living in urban streams based on thermal thresholds relevant to species metabolism, growth, and life history.

**Keywords:** tree canopy cover; water temperature; impervious surface; urban ecology; phenology



## 1. Introduction

Water temperature has long been recognized as an important aquatic environmental variable [1–3] that directly and indirectly affects numerous ecological processes [4–6] and as such is regulated in the United States under the Clean Water Act, Section 303 (d) [7,8]. Increasing water temperature values and variability are known to induce thermal stress in aquatic species that can affect growth, reproductive success, and mortality [9–12]. A recent review of phenology research of aquatic species [13] also identified water temperature as the most important environmental cue for life history behaviors, particularly spawning migration behavior [14–19].

In the last decade, numerous studies have focused on quantifying a stream's thermal regime and drivers of water temperature variability [20–24]. Thermal regime is a term that refers to the stream temperature characteristics and dynamics that we describe based on stream temperature data collected over time [24]. At local spatial scales, important factors that affect stream temperature include riparian vegetation [25], hydrology (e.g., discharge, groundwater source volume, and hyporheic exchange) [1,26,27], and locations where tributaries enter the main channel [1,28]. Local scale variability related to groundwater and tributary connection are relative to baseflow hydrology according to stream size and volume [29,30]. Factors that may affect water temperature variability at the catchment and watershed scales include climate, elevation, and land cover, and geology [1,31,32].

Thermal regimes are sensitive to anthropogenic watershed development that can result in warming and increased variability due to runoff over impervious surfaces and

influence of stormwater pipes [31,33–36]. Previous research has documented the relationship between impervious surface cover and greater incidence and magnitude of stormflow events [37]. These stormflow events can elevate temperatures 3.5 to 7 °C with 3 to 7-h dissipation times, respectively [38]. In addition, extensive subsurface pipe networks, including stormwater pipes, have been added to developed areas that can transfer stormwater with elevated temperatures directly to streams. These pipes can also indirectly interact with groundwater to affect water temperature and baseflow variability, depending on local site conditions. Direct connection of pipes to groundwater can add a constant, stabilizing baseflow from outflow and leaks [33,35].

Land cover within riparian areas and strategic placement of riparian trees can affect shading from solar radiation, heat fluxes in riparian areas, and water temperature variation [2,25,39–41]. Aside from direct shading effects, riparian trees can also create humid microclimates over streams that can stabilize water temperature variability [42,43], with consensus in the literature that riparian tree effects on microclimate generally occur up to about one tree height (15 to 60 m) away from the edge of the stream [39]. Daily maximum water temperature differences between forested and non-forested stream sites can be 4.2–4.9 °C cooler in forested stream reaches [44,45].

While extensive research has been conducted on thermal regulation by riparian tree shading in agricultural and mixed ag-forest watersheds [39,46], little is known about the thermal influence of riparian trees in urban watersheds (>15% impervious land cover) [47–50]. To our knowledge, there is only one research study currently available [49] that has investigated the relationship between tree canopy cover in riparian buffers and water temperature within urban catchments, finding no significant relationship between canopy cover and water temperature. Therefore, the aim of this study was to quantify the effects of riparian and loggershed scale variables on water temperatures in stream sites with 4 to 62% impervious land cover within the "loggershed." We introduce the term "loggershed", which refers to the watershed of the natural and build network draining into each temperature logger point location. We calculated water temperature metric values for each logger location relevant to aquatic species life history, thermal stress, and critical thermal maximum water temperatures to explore relationships between land cover, human development, and potential water temperature changes within the loggershed.

We hypothesized that: (1) The greater the percentage tree canopy within riparian areas along stream networks at the loggershed scale, the lower the frequency and duration of exceedance of water temperature stress threshold values, magnitude of change in water temperature, and variability of water temperature; (2) The greater the percentage impervious surface within riparian areas along stream networks at the loggershed scale, the greater the frequency and duration of exceedance of water temperature stress threshold values, magnitude of change in water temperature, and variability of water temperature; (3) The greater the percentage tree canopy within the loggershed, the lower the frequency and duration of exceedance of water temperature stress threshold values, magnitude of change in water temperature, and variability of water temperature; (4) The greater the percentage impervious surface within the loggershed, the greater the frequency and duration of exceedance of water temperature stress threshold values, magnitude of change in water temperature, and variability of water temperature; (5) Land cover quantified within wider, 30 m riparian areas along both sides of the stream network at the loggershed scale will have a greater effect on water temperature than narrower, 5 m riparian areas; and (6) The greater the length of stormwater pipes (km) in the loggershed, the greater the water temperature variability.

## 2. Materials and Methods

### 2.1. Study Design

This study included 14 stream temperature monitoring sites, defined at what we term the "loggershed" scale (the watershed of the natural and build network draining into each temperature logger point location). For this study we used standard spatial

scale boundaries, as defined by the national Watershed Boundary Dataset for the United States [51,52]. The local scale refers to a stream reach or length of stream; the catchment scale refers to an area defined by the Hydrologic Unit, 12-digit Code (HUC 12) boundary; and the watershed scale refers to the area defined by the Hydrologic Unit, 10-digit Code (HUC 10) boundary [51]. The sites for this study are all within the Beaverdam Run-Lock Raven Reservoir, Red Run-Gwynns Falls, and Dead Run-Gwynns Falls catchments (HUC 12) in the Baltimore, Maryland area (Figure 1). Stream sites were selected from subwatersheds (HUC 12) with USGS gages to estimate base flow from continuous flow data and to download available water temperature data [53,54].

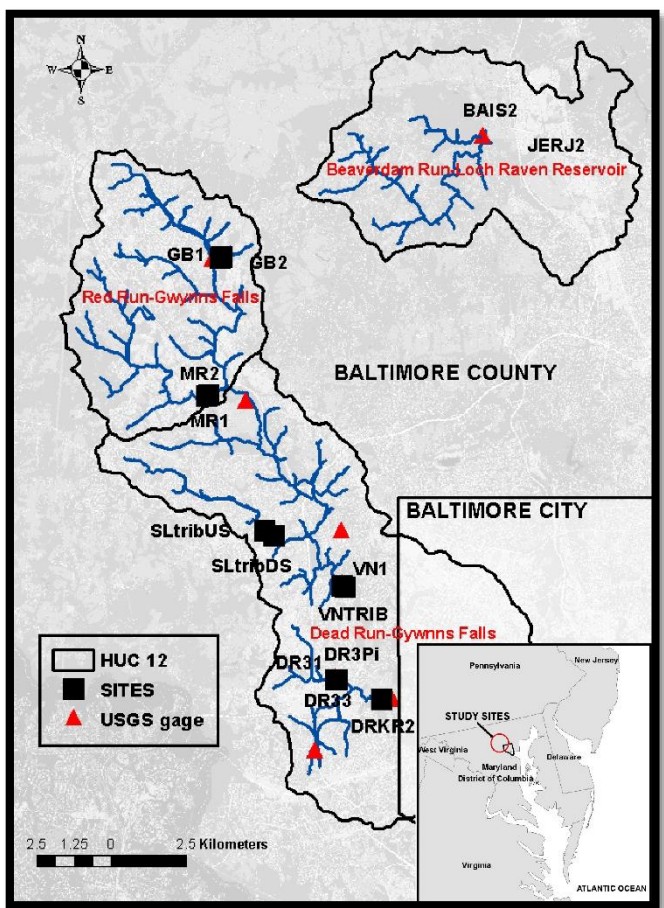

**Figure 1.** Water temperature logger locations near Baltimore, Maryland.

### 2.2. Loggershed and GIS Data

Defining sites at the loggershed scale allowed us to combine geospatial, high resolution (1 m) stream network (USGS, National Map) and land cover data [55,56] for each water temperature logger location. Study site watershed, stream networks, hydrologic, and land cover geospatial data were extracted and derived using ArcMap v.10.3, GIS software, Spatial Analyst application tools (ESRI, 2011). Geospatial land cover and pipes data were extracted at the loggershed scale to investigate the influence of extracted variables on water temperature (Table 1). The choice of variables was based on known significant relationships between the variable and water temperature variability in the literature and the availability of data.

**Table 1.** Loggershed hydrology and pipes variables for study sites.

| Site | Area (km²) | Stream (km) | Pipe (km) | Baseflow | Elevation | Slope | Aspect |
|---|---|---|---|---|---|---|---|
| BAIS2 | 3.78 | 5.24 | 0.63 | 0.0435 | 170.51 | 2.72 | 76.88 |
| DR3-1 | 4.87 | 6.30 | 26.94 | 0.0245 | 132.35 | 1.37 | 86.88 |
| DR3-3 | 4.90 | 6.33 | 27.14 | 0.0246 | 132.23 | 1.38 | 8.54 |
| DR3Pi | 4.88 | 6.32 | 26.94 | 0.0245 | 132.35 | 1.37 | −1.78 |
| DRKR2 | 13.93 | 20.12 | 63.73 | 0.0642 | 130.59 | 1.42 | 10.64 |
| GB1 | 1.42 | 1.84 | 2.51 | 0.0105 | 195.71 | 1.72 | 18.11 |
| GB2 | 1.41 | 1.75 | 2.51 | 0.0105 | 195.84 | 1.72 | 65.16 |
| JERJ2 | 22.82 | 31.55 | 23.05 | 0.2626 | 175.71 | 2.14 | −15.23 |
| MR1 | 17.75 | 24.64 | 70.25 | 0.0013 | 184.44 | 1.83 | −3.81 |
| MR2 | 17.73 | 24.56 | 70.23 | 0.0013 | 184.47 | 1.83 | 86.61 |
| SLtribDS | 8.61 | 9.84 | 63.02 | 0.0441 | 164.51 | 1.30 | 16.24 |
| SLtribUS | 7.98 | 9.34 | 57.41 | 0.0408 | 165.53 | 1.28 | 18.85 |
| VN1 | 81.86 | 116.71 | 402.24 | 0.6401 | 170.04 | 1.72 | 75.42 |
| VNTrib | 81.89 | 116.82 | 402.24 | 0.6404 | 170.02 | 1.72 | 54.32 |

### 2.2.1. Land Cover Data

We calculated total tree canopy and impervious surface area and percent for each study site loggershed using land cover data developed by the University of Vermont Spatial Analysis Lab [55,56]. These data were also extracted to calculate total tree canopy and impervious surface for 5 and 30 m riparian areas on either side of a polyline for the stream network of each loggershed (Figure 2). This high resolution (1 m) raster, 13-class land cover data were downloaded from the Chesapeake Conservancy, Land Cover Data Project 2013/2014 website [57]. The land cover class 3 (Tree Canopy) was extracted to calculate total tree canopy and the land cover classes 7 (Structures), 8 (Impervious Surfaces), and 9 (Impervious Roads) were combined and extracted to calculate total impervious surface area for each loggershed.

### 2.2.2. Hydrology Data

Networks of NHD Plus USGS hydroline data (streams) were extracted geospatially within each loggershed boundary from 1 m resolution raster-based digital elevation models (DEM). Logger site location points were snapped to the closest stream locations, and these were used as pour points for the construction of the DEM for each logger location "loggershed". All hydroline and DEM data were downloaded from The National Map, NHD Plus server [51]. Hydrologic variables calculated for each extracted loggershed stream network included total area of the loggershed (km²), total overall stream length (km), elevation (m), and aspect (as watershed slope direction in degrees; 0–359.9 clockwise starting at due N), and slope (%). Elevation, aspect, and slope were all calculated as a loggershed-based area-weighted average.

### *2.3. Water Temperature Data*
### 2.3.1. Logger Data

A total of 14 water temperature loggers (HoboV2Pro) were calibrated using ice bath methods [58] to an accuracy of $+/- 0.2$ °C and deployed at each site from December 2015 to November 2016. Loggers were shielded from solar radiation using PVC pipe and attached close to the bottom of the stream to a rebar pounded into the stream bed or bank. Water temperature loggers were installed within each stream reach at 50 m intervals. This distance varied to pick up variation at transition zones between tributary confluences and stormwater pipe outfalls at each site. Shaded HoboV2Pro loggers were also deployed at each site to measure on site air temperature. Data were recorded at 15 min intervals for both water and air temperature.

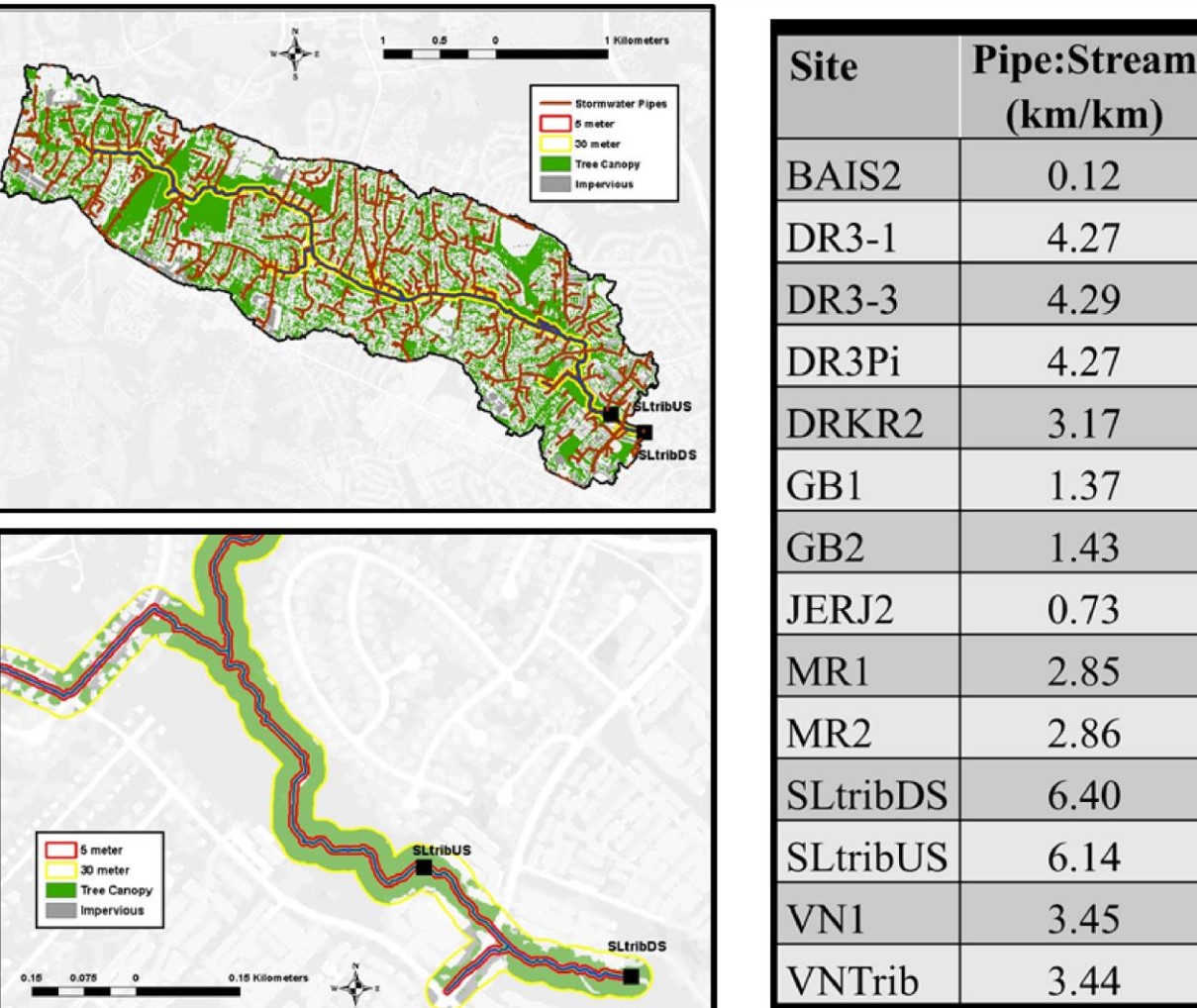

**Figure 2.** Loggershed pipe to stream ratio (km/km) for all sites and example of the SLtribDS site loggershed, showing the length of stormwater pipes and closeup of the tree canopy and impervious land cover within riparian areas (5 and 30 m).

| Site | Pipe:Stream (km/km) |
|------|---------------------|
| BAIS2 | 0.12 |
| DR3-1 | 4.27 |
| DR3-3 | 4.29 |
| DR3Pi | 4.27 |
| DRKR2 | 3.17 |
| GB1 | 1.37 |
| GB2 | 1.43 |
| JERJ2 | 0.73 |
| MR1 | 2.85 |
| MR2 | 2.86 |
| SLtribDS | 6.40 |
| SLtribUS | 6.14 |
| VN1 | 3.45 |
| VNTrib | 3.44 |

2.3.2. Water Temperature Metrics

Mean, maximum, and minimum daily water temperature data were compiled for April to October 2016. For the analysis, we focused on April to October water temperatures, with April to June generally representing the time for fish spawning and growth of early life stages (blacknose dace); July and August as the most likely time of year for aquatic species thermal stress and reduced baseflow; and September to October as a transition period when water temperatures cool and various aquatic species migrate to spawning grounds (brook trout) or move to other habitats to feed [59–61].

Mean, maximum, and minimum daily water temperature data were used to calculate metrics using the StreamThermal software package for R [62] for frequency and duration of exceedance of aquatic species thermal stress water temperatures, magnitude of water temperature change, and water temperature variability. We used thermal stress tolerance [63,64] and critical thermal maximum (CTM) water temperatures as defined by exposure studies for brook trout, rainbow trout, blacknose dace, and virile crayfish that are known to be present in sites for this study (Table 2). The temperature at which an aquatic organism loses equilibrium is known as the CTM water temperature [65,66].

**Table 2.** Thermal stress and critical thermal maximum (CTM) water temperatures for trout, dace, and crayfish species known to occur at sites for this study.

| Scientific Name | Common Name | Temperature | Reference |
|---|---|---|---|
| *Salvelinus fontinalis* | Brook Trout | 21–24 °C (stress) <br> 30 °C (CTM) | [67,68] |
| *Onchorynchus mykiss* | Rainbow Trout | 20 to 22 °C (stress) <br> 29 °C (CTM) | [68,69] |
| *Rhinichthys atratulus* | Blacknose Dace | 29.3 °C (CTM) | [70] |
| *Faxonius virilis* | Virile Crayfish | 25 °C (stress) <br> 26 °C (stress) | [71,72] |

Specifically, frequency of exceedance (FmaxcT; [62]) metric values of the thermal stress threshold water temperatures (# days daily maximum temperature ($n$) $\geq$ threshold water temperature) were calculated for rainbow trout ($\geq$20 °C) [69]; virile crayfish ($\geq$25 °C) [71]; and the CTM threshold for brook trout ($\geq$30 °C) [68]. The maximum number of consecutive days the maximum daily water temperature exceeded threshold values Duration ($n$) [70] were also calculated to quantify how may days of the year aquatic species were exposed to thermal stress (Table 3). Water temperature metrics to quantify magnitude of change included moving average of daily maximum temperature (MovingAMaxT) for 7, 14, and 21 days, and average of daily maximum water temperature per month (Monthly ADMax) [62,73,74]. Water temperature metrics to quantify variability included maximum range per month (greatest value per month for difference between daily maximum and minimum temperature), and variance of the mean daily water temperature for each month [22,62,73,74] (Table 3).

**Table 3.** Water temperature metrics to quantify frequency and duration of exceedance, magnitude of change, and variability.

| Metric | Description | Reference |
|---|---|---|
| Frequency (FmaxcT) | Number days max temperature $\geq$ 20–30 °C | [62] |
| Duration ($n$) | Maximum consecutive days per event $\geq$ 20 °C | [70] |
| MovingAMaxT | Max of 21, 14, 7 day moving average of max | [62] |
| Monthly ADmax | Average daily max, April to October | [62] |
| Max range per month | Greatest value for difference between daily max and daily min temperature per month | [70] |
| Monthly variance | Variance of mean daily temperatures | [22] |

*2.4. Data Analysis*

2.4.1. Correlation Analysis

Pearson correlation analysis was applied to identify relationships between water temperature metric values and predictive variables, using a correlation coefficient threshold $\geq$ 0.7 and $p$ value $\leq$ 0.05 to retain variables for further predictive model analysis. Specifically, Pearson correlation analyses were applied to water temperature metrics (FmaxcT, Events ($n$), Duration ($n$), MaxMovingAMaxT, Monthly ADMax, Max range per month, and Monthly variance) and the following predictive variables for each site: % tree canopy (5 m riparian area, 30 m riparian area, and loggershed); % impervious surface (5 m riparian area, 30 m riparian area, and loggershed); total stormwater pipe length per loggershed (km); stormwater pipe length to stream length ratio; slope, aspect, and baseflow index.

2.4.2. Mixed-Effects Models

Significantly correlated variables were identified and used to test predictive linear mixed-effects models (lme R package) [75], with each metric representing the independent ($x$) variable (metrics listed in Table 3) and the % land cover (percentage in loggershed,

5 m riparian area, 30 m riparian area), stormwater pipe length, and hydrology (slope, aspect, baseflow index) variables representing the dependent (y) variables. Predictive mixed-effects models tested the following hypotheses: (1) Land cover percentages in 30 m riparian areas will have a greater effect on water temperature metrics than land cover percentages in 5 m riparian areas; (2) Land cover percentages at the loggershed scale will have a greater effect on water temperature metrics than land cover percentages within 5 and 30 m riparian areas; (3) Percentage tree canopy cover will have more of an effect than impervious land cover on water temperature metrics within the 5 m riparian area, 30 m riparian area, and loggerhead scales; (4) The total length of stormwater pipes at the loggershed scale will have a greater effect on water temperature variability metrics than tree canopy and impervious land cover at the loggerhead scale.

### 2.4.3. Candidate Model Approach

The percent tree canopy cover and percent impervious cover values were inherently correlated at the loggershed scale within the 30 m riparian areas and within the 5 m riparian areas, because they were geospatial data sets extracted from the same area. Therefore, for the sake of mixed-effects model analysis, we used a candidate model approach where we tested the tree canopy and impervious land cover independent (y) variables separately (only include percent tree canopy cover or percent impervious canopy cover) for each candidate model to test variables related to each water temperature metric dependent variable (x). We used this candidate model approach as well separately for each spatial scale, which resulted in 3 sets of candidate models (6 total) (separated for tree canopy cover and impervious cover) for loggershed, 30 m, and 5 m scales (see Table 4 for an example). Each candidate model used water temperature metric values for the 14 study sites as the x value, testing one metric at a time (6 candidate models per metric) for all metrics listed in Table 3. Candidate models for each metric were retained for comparison and further analysis if the adjusted $R^2$ value was $\geq 0.7$ and the *p* value for all variables was $\leq 0.05$. To select the overall best candidate model from all 6 per metric, we selected the one with the highest $R^2$ value, lowest *p* value, and lowest Akaike's information criterion (AIC) value for goodness of fit [76]. All metric calculations, Pearson correlation analyses, mixed-effects models, and candidate model analyses were completed using R version 4.1.0 [77].

**Table 4.** Mixed-effects model variables to test hypotheses about effects of baseflow, land cover, and stormwater pipes (y variables) on water temperature metrics (x variables). Hypotheses were tested using datasets separated by land cover type (tree canopy or impervious land cover) and spatial scale (loggershed, 30 m, and 5 m) using a candidate model approach.

| Water Temperature Metric (x) | Independent Variables (y) |
|---|---|
| Frequency, Duration | Baseflow + SWPipes + TCLgshd |
| (FmaxcT, Events, *n* days per event) | Baseflow + SWPipes + TC30 |
| Magnitude | Baseflow + SWPipes + TC5m |
| (MaxMovingAMaxT, ADmax) | Baseflow + SWPipes + ImpLgshd |
| Variability | Baseflow + SWPipes + Imp30m |
| (Max range, Variance of mean) | Baseflow + SWPipes + Imp5m |

SWPipes = all stormwater pipes per loggershed; TC = tree canopy; Imp = impervious surface.

## 3. Results

### 3.1. Site Variables and Correlations

#### 3.1.1. Site Characteristics

Site elevation ranged from 132 to 184 m, slope ranged from 1.3 to 2.7, aspect varied from −1.8 to 86.9, and baseflow index ranged from 0.01 to 0.64 (see Table 3 for hydrologic variables). The percent tree canopy cover per loggershed ranged from 23.22 to 79.94% and the percent impervious surface per loggershed ranged from 3.67 to 62.43%. In the 5 m geospatially extracted riparian areas, the percent tree canopy cover ranged from 52.77 to 98.05% and the percent impervious cover ranged from 0.09 to 27.42%. In the 30 m

geospatially extracted riparian areas, the percent tree canopy ranged from 46.09 to 98.01% and the percent impervious cover ranged from 0.26 to 41.78%. The length of stormwater pipes per loggershed ranged from 0.63 to 402.24 km (Figures 2 and 3).

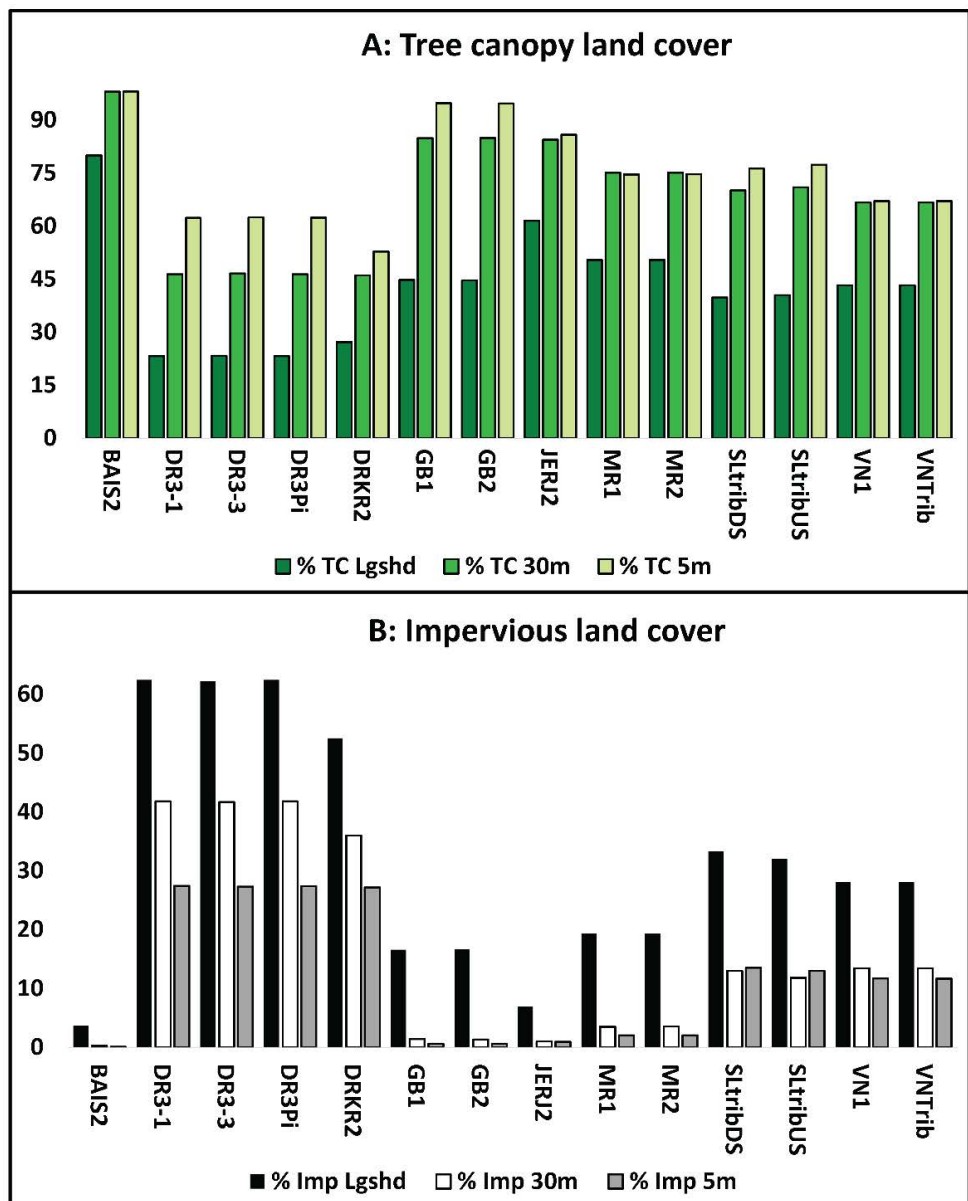

**Figure 3.** Tree canopy land cover percentages (**A**) and impervious land cover percentages (**B**) at study sites. (**A**) shows the percentage tree canopy within the loggershed (% TC Lgshd), within 30 m riparian areas (% TC 30m), and within 5 m riparian areas (% TC 5m). (**B**) shows the percentage impervious within the loggershed (% Imp Lgshd), within 30 m riparian areas (% Imp 30m), and within 5 m riparian areas (% Imp 5m).

### 3.1.2. Overall Variable Correlations

This study identified significant relationships between water temperature, riparian land cover, and total length of stormwater pipes quantified within "loggersheds" with 4 to 62% impervious land cover. Out of the 13 initial variables (Table 1, Figure 3), Pearson correlation analysis identified significant correlations (correlation coefficient threshold $\geq 0.7$ and $p$ value $\leq 0.05$) for the following predictive variables that were retained for further analysis in multiple linear mixed-effects models: six tree canopy and impervious surface land cover variables (5 m riparian area, 30 m riparian area, and loggershed); amount of

stormwater pipes per loggershed; and baseflow index. Of these eight variables, the most frequent number of significant correlations to water temperature metrics were with percent tree canopy in the riparian areas. The percent tree canopy in the 5 m riparian area had 17 significant correlations with water temperature metrics (0.7 to 0.9, $p \leq 0.05$ to 0.00), and the percent tree canopy in the 30 m riparian area had 8 significant correlations to water temperature metrics (0.7 to 0.9, $p \leq 0.01$ to 0.00) (Figure 4).

| Metric | TC5m | P value | TC30m | P value |
|---|---|---|---|---|
| Frequency 20-29°C | -0.70 | 0.00 | | |
| Frequency 25-29°C | -0.90 | 0.00 | | |
| Frequency ≥25°C | -0.80 | 0.00 | | |
| n days ≥20°C | -0.80 | 0.00 | | |
| MaxT21 | -0.75 | 0.02 | | |
| MaxT14 | -0.71 | 0.03 | | |
| MaxT7 | -0.66 | 0.05 | | |
| AprADMax | -0.80 | 0.00 | -0.90 | 0.00 |
| MayADMax | -0.90 | 0.00 | -0.80 | 0.00 |
| JuneADMax | -0.90 | 0.00 | -0.80 | 0.00 |
| JulyADMax | -0.90 | 0.00 | -0.70 | 0.00 |
| AugADMax | -0.70 | 0.01 | | |
| SeptADMax | -0.80 | 0.00 | | |
| OctADMax | -0.90 | 0.00 | -0.90 | 0.00 |
| AprVar | -0.80 | 0.00 | | |
| MayVar | -0.70 | 0.00 | | |
| JuneVar | | | -0.70 | 0.00 |
| OctVar | -0.80 | 0.00 | -0.70 | 0.01 |
| OctRange | | | -0.70 | 0.01 |

| Metric | SW pipes | P value |
|---|---|---|
| MaxT21 | 0.85 | 0.00 |
| MaxT14 | 0.89 | 0.00 |
| MaxT7 | 0.92 | 0.00 |
| AprADMax | 0.70 | 0.00 |
| MayADMax | 0.70 | 0.00 |
| JuneADMax | 0.70 | 0.00 |
| JulyADMax | 0.70 | 0.01 |

**Figure 4.** Significant Pearson correlations between water temperature metrics and percent tree canopy within 5 m riparian areas (TC5m) and 30 m riparian areas (TC30m).

This consistency of significant correlations was not the case for relationships between water temperature and percent impervious cover in the 5 m riparian area that had three significant correlations with water temperature metrics (0.7 to 0.8, $p \leq 0.01$ to 0.00), and the percent impervious surface in the 30 m riparian area that had four significant correlations to water temperature metrics (0.7 to 0.9, $p \leq 0.01$ to 0.00). (correlations with 3 metrics), or percent impervious cover in the 30 m riparian area (correlations with 4 metrics).

These results were unexpected and did not support the hypothesis that the land cover in the 30 m riparian area would affect water temperature to a greater extent compared to the land cover in the 5 m riparian area. In addition, the only loggershed scale variable that had significant correlations with water temperature metrics was the total length of stormwater pipes that had seven significant correlations with water temperature metric values (0.70 to 0.92, $p \leq 0.01$ to 0.00).

### 3.2. Frequency and Duration of Exceedance

#### 3.2.1. Metrics

Using April to October maximum daily temperatures (214 days), our study documented 11 out of 14 sites with events where $n \geq 100$ days exceeded thermal stress water temperatures for rainbow trout and four sites where $n \geq 50$ days exceeded the thermal stress water temperature for virile crayfish. Although this study is based on one year at 14 sites, this means that rainbow trout would experience thermal stress 50–60% of the April to October period (214 days total) at 12 sites and virile crayfish would experience thermal stress 20–30% of the time frame at 6 sites.

The total number of days that the daily maximum temperature exceeded the thermal stress water temperature (FmaxT) for rainbow trout ($\geq$20 °C) [73] ranged from 76 to 142 days, and 0 to 80 days for virile crayfish ($\geq$25 °C) [71] at study sites. In addition, the total number of days (FmaxT) that the daily maximum temperature exceeded the critical thermal maximum water temperature for brook trout ($\geq$30 °C) [68] was 1 at the SLtribUS site and 17 at the VNTrib site. The maximum number of consecutive days MaxT was $\geq$20 °C; Duration (*n*) ranged from 59 days in a row to 124 days in row at study sites for the period of April to October (214 days).

### 3.2.2. Significant Correlations and Best Fit Mixed-Effects Models

Pearson correlation analysis identified negative correlations between the percent tree canopy cover in the 5 m riparian area and FmaxcT, 20–29 °C ($-0.70$, $p < 0.00$), FmaxcT, 25–29 °C ($-0.90$, $p < 0.00$), FmaxcT $\geq$ 25 °C ($-0.80$, $p < 0.00$), and Duration (*n*) $\geq$ 20 °C ($-0.80$, $p < 0.00$) (Figure 4). The best model to predict FmaxcT (25–29 °C) included percent tree canopy in the 5 m riparian area (TC5m) ($p = 0.00$, Adjusted $R^2 = 0.71$); the best model to predict FmaxcT ($\geq$25 °C) included percent tree canopy in the 5 m riparian area (TC5m) and baseflow index (BF) ($p = 0.00$, Adjusted $R^2 = 0.73$); and the best model to predict Duration (*n*) of days $\geq$ 20 °C included percent tree canopy in the 5 m riparian area (TC5m), stormwater pipes (SW Pipes), and baseflow index (BF) ($p = 0.00$, Adjusted $R^2 = 0.73$) (Table 5).

**Table 5.** Mixed-effects models to predict water temperature metrics based on April to October 2016 data for 14 study sites. The selected metrics are designed to quantify frequency and duration of exceedance of thermal stress thresholds and magnitude of thermal change for aquatic species in urban streams with 4 to 62% impervious land cover. The models listed in this table are the final, best fit model to predict each metric after testing 6 candidate models per metric. The best fit predictive models for frequency of exceedance of thermal stress temperatures include baseflow (BF) and % tree canopy in 5 m riparian areas (TC5m) variables. The best fit predictive model for duration (Max days $\geq$ 20 °C) included BF, TC5m, and total length of stormwater pipes per loggershed (SWPipes). The best fit predictive models for magnitude of change (MaxT21, MaxT14, MaxT7) included TC5m and SWPipes.

| Metric | Model (Y) | t Value | Pr(>\|t\|) | Model |
|---|---|---|---|---|
| Frequency 20–29 °C | Intercept | 7.54 | 0.00 | 25–29 °C = 139.97–5.73(TC5m) |
| | TC5m | −5.73 | 0.00 | Adjusted R-squared: 0.71 |
| | | | | Model: F statistic 32.79, *p*-value: 0.00 |
| Frequency $\geq$ 25 °C | Intercept | 6.35 | 0.00 | 25–41 °C = 131.20–4.99(TC5m) + 2.37(BF) |
| | TC5m | −4.99 | 0.00 | Adjusted R-squared: 0.73 |
| | BF | 2.37 | 0.04 | Model: F statistic 18.40, *p*-value: 0.00 |
| Duration (*n*) days $\geq$ 20 °C | Intercept | 8.62 | 0.00 | Max Days = 182.20–4.93(TC5m) + 2.33(SWPipes) + 1.51(BF) |
| | TC5m | −4.93 | 0.00 | Adjusted R-squared: 0.73 |
| | SWPipes | 2.33 | 0.04 | Model: F statistic 12.40, *p*-value: 0.00 |
| | BF | 1.51 | 0.16 | |
| MaxT21 | Intercept | 45.67 | 0.00 | MaxT21 = 21.6586–0.0528(TC5m) + 0.0084(SWPipes) |
| | TC5m | −8.85 | 0.00 | Adjusted R-squared: 0.97 |
| | SWPipes | 11.41 | 0.00 | Model *p*-value: 0.00 |
| MaxT14 | Intercept | 78.67 | 0.00 | MaxT14 = 21.7167–0.051(TC5m) + 0.0097(SWPipes) |
| | TC5m | −14.68 | 0.00 | Adjusted R-squared: 0.99 |
| | SWPipes | 22.74 | 0.00 | Model *p*-value: 0.00 |
| MaxT7 | Intercept | 134.11 | 0.00 | MaxT7 = 23.1676–0.0505(TC5m) + 0.0117(SWPipes) |
| | TC5m | −23.26 | 0.00 | Adjusted R-squared: 0.99 |
| | SWPipes | 43.68 | 0.00 | Model *p*-value: 0.00 |

### 3.2.3. Mixed-Effects Model Interpretation

The mixed-effects model for FmaxcT (25–29 °C) predicts that the percent tree canopy in the 5 m riparian area will reduce MaxT water temperature values by 5.7 °C and 5.0 °C for the mixed-effects model to predict FmaxcT ($\geq$25 °C). The mixed-effects model for Duration

($n$) of days $\geq 20\ °C$ predicts that the total length of stormwater pipes in the loggershed and baseflow will increase the duration of $\geq 20\ °C$ days by 2; the percent tree canopy in the 5 m riparian area will reduce the duration of $\geq 20\ °C$ days by 5. Therefore, the prediction that the greater the tree canopy per surface in the riparian area, the lower the frequency and duration of exceedance for water temperature stress thresholds was well supported by the data.

### 3.3. Annual Magnitude of Change
#### 3.3.1. Metrics

The moving average of daily maximum temperature (MovingAMaxT) ranged from 16 to 22 $°C$ for 21 days, and 17 to 22 $°C$ for 14 days, which both include values for sites that exceed the stress threshold water temperature value for rainbow trout ($\geq 20\ °C$) [73]. The 7d Moving AMaxT range of values, 18 to 25 $°C$, includes 10 sites that exceed the stress threshold water temperature value for rainbow trout and one site that exceeds the stress threshold water temperature value for virile crayfish ($\geq 25\ °C$) [71].

#### 3.3.2. Significant Correlations and Best Fit Mixed-Effects Models

Pearson correlation analysis identified negative correlations between the percent tree canopy cover in the 5 m riparian area and MaxT21 ($-0.75$, $p < 0.02$), MaxT14 ($-0.71$, $p < 0.03$), and MaxT7 ($-0.66$, $p < 0.05$). Pearson correlation analysis also identified positive correlations between the total stormwater pipe length in the loggershed and MaxT21 (0.85, $p < 0.00$), MaxT14 (0.89, $p < 0.00$), and MaxT7 (0.92, $p < 0.00$) (Figure 4). The best model to predict MaxT21 ($p = 0.00$, Adjusted $R^2 = 0.97$), MaxT14 ($p = 0.00$, Adjusted $R^2 = 0.99$), and MaxT7 ($p = 0.00$, Adjusted $R^2 = 0.99$) all included percent tree canopy in the 5 m riparian area (TC5m) and total length of stormwater pipes at the loggershed scale (SWpipes).

#### 3.3.3. Mixed-Effects Model Interpretation

These models for MaxT21, MaxT14, and MaxT7 predict that the percent tree canopy in the 5 m riparian area will reduce MaxT water temperature values by ~0.05 $°C$; the length of stormwater pipe in the loggershed will increase the MaxT water temperature values by 0.01 $°C$ (Table 5). The Moving AMaxT range of values exceeded the stress threshold water temperature for rainbow trout at 10 sites and for virile crayfish at one site. Significant mixed-effects models for MovingAMaxT predicted that the percent tree canopy in the 5 m riparian area would reduce the MaxT water temperature by ~0.05 $°C$. In addition, significant mixed-effects models for May–August ADMax water temperature predicted that the percent canopy in the 5 m riparian area would reduce the MaxT water temperatures ~ 4–7 $°C$. Therefore, the prediction that the greater the tree canopy in the riparian area, the lower the water temperature was well supported by the data.

### 3.4. Monthly Magnitude of Change
#### 3.4.1. Metrics

The mean Monthly ADMax (April to October) for all sites ranged from 15 $°C$ in April to 25 $°C$ in August. The JuneADmax and SeptemberADmax values for 11 out of 14 sites were $\geq 20\ °C$ (stress threshold water temperature for rainbow trout). The JulyADMax and AugustADMax values were $\geq 20\ °C$ at all sites; and the April, May, and October ADMax values were all $\leq 20\ °C$ at all sites.

#### 3.4.2. Significant Correlations and Best Fit Mixed-Effects Models

Pearson correlation analysis identified negative correlations between the percent tree canopy cover in the 5 m riparian area and all monthly ADmax values (April to October), (correlation range $-0.70$ to $-0.90$, $p$ values range $< 0.00$ to $0.01$. Pearson correlation analysis also identified correlations between the percent tree canopy cover in the 30 m riparian area and monthly ADmax values for April to July and October (correlation range $-0.70$ to $-0.90$, $p$ values range $< 0.00$ to $0.00$). There were also positive correlations between percent

impervious surface in the 30 m riparian area and both April ADmax (0.8, $p < 0.00$) and May ADmax (0.7, $p < 0.01$). The total length of stormwater pipe in the loggershed was positively correlated to April through July ADMax (0.70, $p$ values range $< 0.00$ to 0.01) (Figure 4).

The best model to predict April ADmax ($p = 0.00$, Adjusted $R^2 = 0.79$) included percent impervious surface in the 30 m riparian area (Imp30m) and total length of stormwater pipes at the loggershed scale (SWpipes). The best model to predict May ADmax ($p = 0.00$, Adjusted $R^2 = 0.89$) included percent tree canopy in the 5 m riparian area and total length of stormwater pipes at the loggershed scale. The best models to predict June ADmax ($p = 0.00$, Adjusted $R^2 = 0.88$) and July ADmax ($p = 0.00$, Adjusted $R^2 = 0.83$) included percent tree canopy in the 5 m riparian area, total length of stormwater pipes at the loggershed scale, and baseflow index. The best model to predict August ADmax ($p = 0.00$, Adjusted $R^2 = 0.68$) included percent tree canopy in the 5 m riparian area and baseflow index. The best model to predict October ADmax ($p = 0.00$, Adjusted $R^2 = 0.88$) included percent tree canopy in the 30 m riparian area and baseflow index (Table 6).

**Table 6.** Mixed-effects models to predict water temperature metrics for magnitude of change and variability based on April to October 2016 data for 14 study sites. The average of daily maximum water temperature per month (Monthly ADMax), maximum range per month, and variance of the mean daily water temperature per month are designed to quantify the magnitude and variability in water temperature change per month for aquatic species in urban streams with 4 to 62% impervious land cover. The models listed in this table are the final, best fit model to predict each metric after testing 6 candidate models per metric. The predictive variables included in the best fit predictive models for monthly ADMax are variable by month, suggesting that sources of variability in urban stream habitats may affect aquatic species differently during times of spawning (April to May), growth of early life stages (June), and times typically most thermally stressful (July to August). Final candidate models to predict water temperature variability were only significant for June variance and October range.

| Metric | Model (Y) | t Value | Pr(>|t|) | Model |
|---|---|---|---|---|
| AprADMax | Intercept | 91.95 | 0.00 | AprADMax = 14.23 + 4.39(Imp30m) + 2.61(SWPipes) |
| | Imp30m | 4.39 | 0.00 | Adjusted R-squared: 0.79 |
| | SWPipes | 2.61 | 0.02 | Model: F statistic 25.48, *p*-value: 0.00 |
| MayADMax | Intercept | 27.28 | 0.00 | MayADmax = 19.78−6.72(TC5m) + 2.43(SWPipes) |
| | TC5m | −6.72 | 0.00 | Adjusted R-squared: 0.89 |
| | SWPipes | 2.43 | 0.03 | Model: F statistic 54.86, *p*-value: 0.00 |
| JuneADMax | Intercept | 19.40 | 0.00 | JuneADmax = 25.94−5.28(TC5m) + 2.63(SWPipes) + 2.08(BF) |
| | TC5m | −5.28 | 0.00 | Adjusted R-squared: 0.88 |
| | SWPipes | 2.63 | 0.03 | Model: F statistic 31.79, *p*-value: 0.00 |
| | BF | 2.08 | 0.06 | |
| JulyADMax | Intercept | 17.56 | 0.00 | JulyADmax = 28.73−4.25(TC5m) + 2.28(SW Pipes) + 1.75(BF) |
| | TC5m | −4.25 | 0.00 | Adjusted R-squared: 0.83 |
| | SWPipes | 2.28 | 0.05 | Model: F statistic 21.58, *p*-value: 0.00 |
| | BF | 1.75 | 0.11 | |
| AugADMax | Intercept | 15.40 | 0.00 | AugADmax = 30.87−3.64(TC5m) + 3.22(BF) |
| | TC5m | −3.64 | 0.00 | Adjusted R-squared: 0.68 |
| | BF | 3.22 | 0.01 | Model: F statistic 14.72, *p*-value: 0.00 |
| OctADMax | Intercept | 92.43 | 0.00 | OctADmax = 17.26−9.43(TC30m) + 2.36(BF) |
| | TC30m | −9.43 | 0.00 | Adjusted R-squared: 0.88 |
| | BF | 2.36 | 0.04 | Model: F statistic 45.89, *p*-value: 0.00 |
| JuneVar | Intercept | 10.34 | 0.00 | JuneVar = 0.89 + 5.38(Imp5m) + 3.58(SWPipes) |
| | Imp5m | 5.38 | 0.00 | Adjusted R-squared: 0.79 |
| | SWPipes | 3.58 | 0.00 | Model: F statistic 27.4, *p*-value: 0.00 |
| OctRange | Intercept | 15.35 | 0.00 | OctRange = 6.37− 5.15(TC) |
| | TC | −5.15 | 0.00 | Adjusted R-squared: 0.70 |
| | | | | Model: F statistic 26.56, *p*-value: 0.00 |

### 3.4.3. Mixed-Effects Model Interpretation

These models for monthly ADmax predict that the percent tree canopy in the 5 m riparian area will decrease average daily maximum water temperature per month by ~6.7 °C in May, ~5.3 °C in June, ~4.3 °C in July, and ~3.6 °C in August. The models predict that the total length of stormwater pipes per loggershed will increase the average daily maximum water temperature per month by ~2.6 °C in April, ~2.4 °C in May, ~2.6 °C in June, and ~2.3 °C in July. The models predict that baseflow will increase the average daily maximum water temperature per month by ~2.1 °C in June, ~1.8 °C in July, ~3.2 °C in August, and ~2.4 °C in October. In addition, the model for April ADmax predicts that the percent impervious surface cover in the 30 m riparian area will increase the average daily maximum water temperature per month by ~4.4 °C; the model for October ADmax predicts that the percent tree canopy cover in the 30 m riparian area will decrease the average daily maximum water temperature per month by ~9.4 °C (Table 6).

### 3.5. Monthly Variability

#### 3.5.1. Metrics

The mean maximum range per month (April to October) for all sites ranged from 3.46 in September to 7.52 in April. The greatest variance of the mean daily water temperature per month for all sites ranged from 1.4 in June to 7.4 in May. The greatest values for mean maximum range per month and variance of the mean were both in April and May (Figure 5).

#### 3.5.2. Significant Correlations and Best Fit Mixed-Effects Models

Pearson correlation analysis identified negative correlations between the percent tree canopy cover in the 5 m riparian area and April ($-0.80$, $p < 0.0$), May ($-0.70$, $p < 0.00$), and October ($-0.80$, $p < 0.00$) variance of the mean daily water temperature. June ($-0.70$, $p < 0.00$) and October ($-0.70$, $p < 0.01$) variance of the mean were negatively correlated to the percent tree canopy cover in the 30 m riparian area. Positive correlations to impervious surface cover included correlations of April variance to percent impervious cover in the 5 m riparian area ($0.70$, $p < 0.01$), and May variance to impervious surface cover in the 5 m riparian area ($0.80$, $p < 0.00$) and percent impervious cover in the 30 m riparian area ($0.90$, $p < 0.00$). October maximum range of water temperature was the only maximum range per month value with significant correlations. These correlations included a negative correlation with percent tree canopy cover in the 30 m riparian area ($-0.70$, $p < 0.01$), a positive correlation with percent impervious cover in the 5 m riparian area ($0.70$, $p < 0.00$), and a positive correlation with percent impervious cover in the 30 m riparian area ($0.70$, $p < 0.00$) (Figure 4).

The only mixed-effects linear regression models with adjusted $R^2 \geq 0.7$ and $p < 0.05$ were models to predict June variance and October maximum range that included percent impervious surface area in the 5 m riparian area, total length of stormwater pipes per loggershed, and percent tree canopy cover per loggershed.

#### 3.5.3. Mixed-Effects Model Interpretation

The best model for June variance ($p = 0.00$, Adjusted $R^2 = 0.79$) predicts that the percent impervious surface area in the 5 m riparian area will increase monthly variance by ~5.4 °C and the total length of stormwater pipes per loggershed will increase monthly variance by ~3.6 °C. The best model for October maximum range ($p = 0.00$, Adjusted $R^2 = 0.70$) predicts that the percent tree canopy cover per loggershed decreases the monthly variance by ~5.2 °C (Table 6).

The prediction that the amount of impervious surface in the loggershed the greater the water temperature variability was not well supported by the data, as only the June variance predictive model included this variable in the candidate model with a lower adjusted $r^2$ (0.71) compared to the other candidate models (0.77 and 0.79). Otherwise, we found no significant mixed effects models to predict frequency, duration, magnitude of change,

and variability that included this variable. The prediction that the greater the length of stormwater pipes in the loggershed, the greater the water temperature variability was not well supported by the data either, as there were no significant correlations between any monthly variance or maximum range values, and only the mixed effects model for June variance included stormwater pipes as a predictive variable.

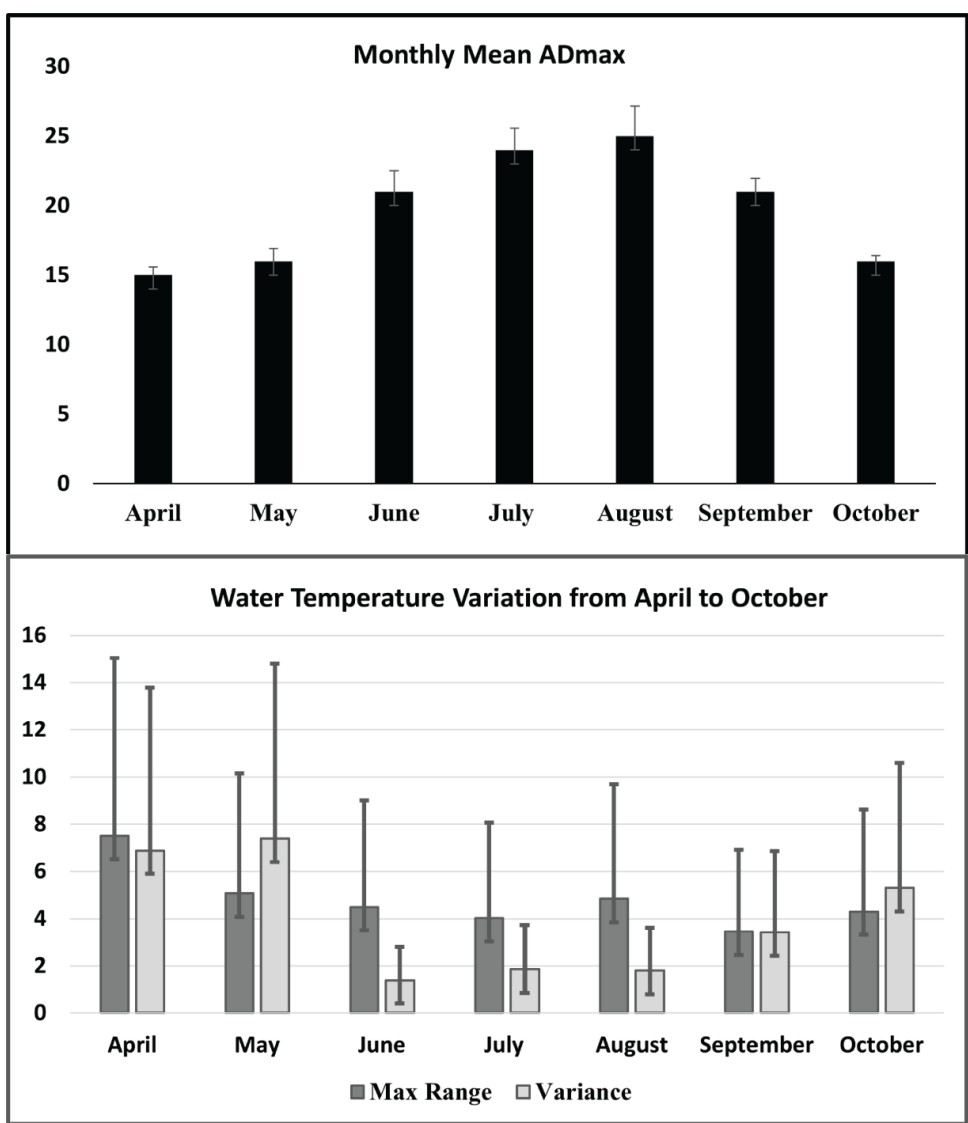

**Figure 5.** Monthly magnitude of change and variability per month for April to October. The monthly mean ADMax, maximum range of water temperature per month, and variance of mean daily water temperature per month and standard deviation values are shown.

## 4. Discussion

To our knowledge, this is the first time that the relationship between stormwater pipes and water temperature metrics has been explored to better understand thermal dynamics in urban watersheds. The results highlight important aspects of thermal habitat quality and water temperature variability for aquatic species living in urban streams based on thermal thresholds relevant to species metabolism, growth, and life history [9,10,78]. Although this study was based on a limited number of study sites from one year of logger data, results highlighted significant negative correlations (19 metrics) between percent of tree canopy in the 5 m riparian area and water temperature.

Our results were consistent with the literature showing a negative effect of urbanization on aquatic organisms, especially sensitive species such as trout, due to thermal stress

and unsuitable thermal habitat [79,80]. Spring to early summer is an important period for spawning behavior for many fish species, including blacknose dace [81] and rainbow trout [82] that occur at sites of this study. Increases in water temperature variability during these months associated with impervious surface cover may result in changes in timing of spawning or changes in emergence and length of adult life stages for aquatic macroinvertebrates [83,84]. Additionally, early life stages of fish that typically develop during the spring and early summer require available prey at the right time for survival and growth into the adult stage [16,17,59,60]. Change in the thermal triggers associated with spawning or emergence can result in seasonal asynchronicity (match mismatch hypothesis) (e.g., [85]). Using water temperature metrics to monitor thermal habitat conditions and to identify times of the year when life history behaviors, such as timing of spawning migrations may be affected, can be a useful tool for managers to better identify the thermal habitat being impaired [22,62,74,86]. Our results showed that such as ADMax values for the months of April, May, and June could be used to monitor the effects of thermal stress and water temperature variation on early life stages of fish.

*Urbanization and Thermal Degradation Mitigation*

Although significant relationships between landcover and thermal metrics were observed, this consistency of significant correlations was not the case for relationships between water temperature and percent tree canopy in the 30 m riparian area (8 metrics), percent impervious cover in the 5 m (3 metrics), or percent impervious cover in the 30 m riparian area (4 metrics). In addition, tree canopy cover at the loggershed scale was the only significant variable retained in the mixed effects model to predict October variability. These results highlighted the importance of investigating other variables that can offset the benefits of riparian trees and influence thermal results in urban streams.

Moreover, this study only found significant correlations between land cover variables in the 5 m and 30 m riparian areas with the April, May, June and October variance and October maximum range water temperature metrics. The only significant candidate mixed effects models for monthly variability were for June, which included impervious surface in the 5 m riparian area and length of stormwater pipes; October maximum range included tree canopy cover at the loggershed scale.

This study showed the greatest values in monthly variance and maximum range in April, May, and June (Figure 5). This variability in April and May was positively correlated to impervious surface in the 5 m and 30 m riparian area. In addition, the June variability was positively correlated to impervious surface in the 30 m riparian area and significant mixed-effects models included impervious surface in the 5 m riparian area and total length of stormwater pipes in the loggershed. These results are consistent with previous research that documented greater fluctuation in water temperatures in urban streams relative to percent impervious surface cover [87–90]. However, in addition to effects of impervious surface cover on water temperature variability, this study identified significant negative correlations between water temperature variability and tree canopy cover. Specifically, this study identified significant negative correlations between April and May water temperature variability and percent tree canopy cover in the 5 m riparian area and between variability in June and percent tree canopy cover in the 30 m riparian area. Results of this study also predicted a reducing effect of tree canopy cover in 5 m riparian areas on May and June water temperatures of ~5–7 °C.

The study found positive significant correlations between the total length of stormwater pipes at the loggershed scale and 8 magnitude water temperature metrics, in loggersheds with up to 5.34 km of pipe to 0.48 km of stream (Figure 2). Although significant mixed effects models to predict water temperature variability included length of stormwater pipes, this result is based on geospatially extracted data and was not based on direct measurement of stormwater pipe leaks, stormwater inflows, green infrastructure effectiveness for treating thermal pollution from stormwater effects, or groundwater input at study sites.

Variability of water temperature is known to be influenced by local, reach scale inputs such as groundwater inputs [1,27] and locations where tributaries enter the main channel [1,28].

Our results showed that the presence of stormwater pipes could potentially offset the benefit of riparian trees, which highlights the need for further investigation of other important variables which may affect riparian restoration outcomes. Similarly, other underground structures could leak water with different temperatures (e.g., cooler), and mask the effect of features such as impervious surfaces that usually cause thermal degradation. Our results highlighted the importance of understanding connections between specific urban features and thermal regimes. This is especially important for a project aiming to mitigate the impact of urban thermal degradation. Additionally, our studies provided insight on how the scale of the study could also influence the scale of the restoration action and outcomes, as some variables may have a more significant influence over larger scales (e.g., impervious surface). In contrast, others might have a more localized impact within the loggershed space (e.g.,) tree canopy. This study highlighted that urban streams are a complex mosaic of intertwining variables that ultimately influence the thermal regime. More research on the thermal sensitivity for each variable is needed in order to develop more meaningful management and mitigation recommendations for thermal degradation.

## 5. Conclusions

The most commonly applied thermal cooling best management practice (BMP) is riparian tree planting, a strategy that has been applied in the United States since the 1970s to mitigate the impacts from logging and agriculture [46,91–93]. Considering the results of this study, which found significant correlations and predicted effects of stormwater pipes, impervious surface, and riparian tree canopy cover on water temperature, further research is needed to identify additional urban variables of importance and if riparian tree canopy cover could still be used to mitigate increases in water temperature in urban catchments [50,94,95]. Future studies using multiple years of water temperature data at the loggershed scale and on-the-ground surveys of tree species within riparian area widths 5 to 30 m and greater, in addition to geospatial data, would be helpful to further investigate characteristics of riparian areas that affect thermal regime along urban streams.

In urban areas with stormwater pipe networks such as the sites in this study with 10 out of 14 sites, $\geq 4$ to 6 km of pipes per 1 km of stream, the local influence of stormwater pipe outlets on water temperature is likely [96]. However, our study design was set up with a logger at every ~50 m and geospatial data were extracted at the loggershed scale. This study design was not set up to effectively quantify the effects of every tributary and stormwater pipe outlet, which would require a more extensive network of loggers upstream and downstream of each tributary confluence and stormwater pipe input longitudinally throughout the stream network. In addition, the baseflow index was estimated using USGS gage watershed locations at a scale too large to quantify local groundwater inputs effectively [97]. Riparian vegetation is also known to influence local variability [25], and local reach scale, transect based surveys of tree canopy cover, solar radiation, and tree species influence on local hydrology and microclimate may offer insights into local water temperature variability [42,98].

Our results confirmed that one of the greatest impacts of urbanization for aquatic species is the induced thermal stress. The extent of thermal stress for aquatic species depends on the availability of habitat with temperatures below thermal stress temperature thresholds and the ability of species to disperse to those habitats. This highlight the importance of conservation and creating coldwater refugia within connected stream networks that would offer refuge during times of thermal stress [28,99–101]. Previous studies have documented coldwater refugia in highly urbanized watersheds (17% impervious) where groundwater enters the stream [102,103]. Further direct measurement of groundwater availability within stream networks at the loggershed scale and catchment (HUC 12) scale may offer opportunities to identify other urban thermal refugia locations, keeping in mind that groundwater can be an important source of water temperature variability at the local

reach scale. Understanding how managers can use cold water refuges to create a mosaic of thermal habitat for fish to thermoregulate may help prevent urban areas from becoming thermal barriers to dispersal throughout stream networks.

**Author Contributions:** Conceptualization, A.T., V.O., M.D.; methodology, A.T., V.O., M.D.; formal analysis, A.T.; investigation, A.T., V.O., M.D.; resources, A.T.; data curation, A.T., V.O.; writing—original draft preparation, A.T.; writing—review and editing, A.T., V.O., M.D.; project administration, A.T., V.O., M.D.; funding acquisition, A.T., V.O., M.D. All authors have read and agreed to the published version of the manuscript.

**Funding:** This research was funded by the USDA Forest Service, Northern Research Station and Fonds de Recherche du Québec—Nature et Technologies (FRQNT) for V. Ouellet.

**Institutional Review Board Statement:** Not applicable.

**Data Availability Statement:** The data presented in this study are available on request from the corresponding author.

**Acknowledgments:** We thank Nancy Sonti, Sarah Hines, Ian Yesilonis, and Ken Belt for field data collection support. We thank Charlie Dow, Stroud Water Research Center; Krista Heinlen, USDA Forest Service, Northern Research Station; and Jarlath O'Neil-Dunne, University of Vermont for help with GIS data processing. We thank Dexter Locke for help with GIS analysis and helpful comments to improve this manuscript.

**Conflicts of Interest:** The authors declare no conflict of interest.

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
