# Peer review of "Riparian Land Cover, Water Temperature Variability, and Thermal Stress for Aquatic Species in Urban Streams"

_water, doi:10.3390/w13192732_

Round 1
Reviewer 1 Report
General comments:
The article presents a significant original finding for the journal, in both topic and scope. The paper is well-organized, with the various properly separated sections. The title is representative of the article contents. The objectives are well articulated. The state of the art is well described and the knowledge gap is clearly defined. All sections are well-developed. The applied research methodology is solid, clear and complete enough to be reproducible. The results are clearly and completely described in an organized manner. The conclusions are justified. The authors have made an important contribution to the field of study. The authors have effectively explained their results in the context of the literature
A list of specific remarks regarding weaknesses and concerns about the manuscript:
Table 5 Mixed-effects models for frequency, duration, and magnitude of change water temperature metrics for all April to October 2016 data. What does it add to the subject area compared with other published material? The sampling in analytical and the validity of questions is prominently in the paper. This statement should give a brief account of the purpose, need, and significance of the investigation. Improve it.
Table 6 Mixed-effects models for the monthly magnitude of change and variability water temperature metrics for the months of April to October 2016. Decide how well the article presents the evidence that supports its argument. You must demonstrate that resilience to extreme temperature events. Do the results and discussion sections clearly summarize and interpret the data?
You might observe that subjects in environ study didn’t accurately represent a diverse population. In Figure 3. you showed impervious and tree canopy land cover percentages within study site loggerheads (% Imp and % TC Lgshd), within 30-meter riparian areas (% Imp 30m 253 and % TC 30m), and within 5-meter riparian areas (% Imp 5m and % TC 5m). Are these methods an appropriate, reasonable means of solving the problem? This point is only valid if researchers can point to recent developments in data gathering.
In Figure 5. you showed the monthly magnitude of change and variability per month for April to October. The monthly mean ADMax, a maximum range of water temperature per month, and variance of mean daily water temperature per month and standard deviation values are shown. The results seem plausible in this part. Methods are not detailed enough. Once described, they should evaluate the trends observed and explain the significance of the results to a wider understanding.
Constructive feedback:
Authors clearly outline the methodology and the expected result, but in the methodology section, you should more elucidate about. I have tried below to suggest some specific questions that, if answered, could illuminate both the message and the benefits of the paper. The statistical and analytical methods are presented in a complete and clear format. However, small issues need further explanation. The conclusions and data interpretation are robust, valid and reliable. The authors have extended their conclusions beyond what their data actually shows, and the potential limitations of the study have been adequately discussed. However, the authors have no proposed relevant and reasonable recommendations based on their results. They should start by describing in simple terms what the data show
The study has helped to advance knowledge, but the novel results have not been appropriately highlighted. You should focus on some innovation in this manuscript. Please try to find literature elaborating river valley and ecological connection. This statement should give a brief account of the purpose, need, and significance of the investigation for instance ecosystem service. Improve it.
The precision of process data and regularity of sampling in time-dependent studies is likely to be a major flaw. Please give more details about the impact of urban patterns on aquatic ecosystems. Are enough examples provided to assist readers in relating to the author?
You mentioned “Further research needed.- The most commonly applied thermal cooling best management practice (BMP) is riparian tree planting, a strategy that has been applied in the United States since the 1970s to mitigate the impacts from logging and agriculture [46,82– 84]. What are its implications for an understanding of scientific results by the general, literate public? In studies carried out over time are there sufficient data points to support the trends described by the authors?
You reported that: “Our results were partially consistent with a previous study by Roy et al. (2005) that quantified tree canopy cover in riparian areas within 30-m widths along urban streams (≥ 15% impervious) that found no significant relationships to water temperature. The Roy et al. (2005) study was based on data collected in the field with riparian area 30-meter widths along 200-m reaches, whereas our study used geospatially extracted 5-meter and 30-meter riparian area data at the loggershed scale”. You might add a chapter that describes a thermal habitat that restricts patterns of occurrence in multiple life-stages of a headwater for trout, dace, and crayfish species. How well does the modelling time-dependent effects of thermal stress on ectotherm growth rates summarize the article? The problem addresses, its techniques, results, and significance? Please specify. Remember that all of your ideas must support your central design.
You might examine that qualitative research, with sufficient descriptive elements and appropriate quotes from interviews unless strong arguments for such repetition are made. Add the impact of buffer zone or morphology or geomorphology of the riverbank in a river valley in an urban area. How the article relates to other work on the same subject? Are they supported by evidence and analysis for example River Habitat assessment in an urbanised river?
Sufficient use of control experiments in detailed methodology and the data analysis is done systematically (in qualitative research) it's all about the information gathered. In this paragraph, you might identify the question, how well the article resolves its central problem.
A brief summary of the paper:
The research is both original and relevant. The paper is well-written with interesting data and cogent arguments. The reporting of data and methodology are sufficiently detailed and transparent to enable reproducing the results. The article is organized logically and easy to follow. The conclusion summarizes the finding and offers suggestions for future research.
Author Response
Water-1352243 R1, Major Revisions Response
Title: Riparian land cover, water temperature variability, and thermal stress for aquatic species in urban streams
Submitted by:
Dr. Anne Timm, USDA Forest Service, Northern Research Station, Baltimore, Maryland, United States
Dr. Valerie Ouellet and Dr. Melinda Daniels, Stroud Water Research Center, Avondale, Pennsylvania, United States
Response to Reviewers
The authors greatly appreciate the feedback from the reviewers. We have carefully considered each comment and changed the manuscript accordingly. Below, we have responded to general comments and then individual, line by line comments from each reviewer. We have indicated what changes have been made in the manuscript for each comment. The latest version of the manuscript with edits has been uploaded.
GENERAL COMMENTS:
Reviewer #1
Comment:
The article presents a significant original finding for the journal, in both topic and scope. The paper is well-organized, with the various properly separated sections. The title is representative of the article contents. The objectives are well articulated. The state of the art is well described and the knowledge gap is clearly defined. All sections are well-developed. The applied research methodology is solid, clear and complete enough to be reproducible. The results are clearly and completely described in an organized manner. The conclusions are justified. The authors have made an important contribution to the field of study. The authors have effectively explained their results in the context of the literature
A brief summary of the paper:
The research is both original and relevant. The paper is well-written with interesting data and cogent arguments. The reporting of data and methodology are sufficiently detailed and transparent to enable reproducing the results. The article is organized logically and easy to follow. The conclusion summarizes the finding and offers suggestions for future research.
Response-The authors appreciate the acknowledgements from the reviewer that the topic is significant and original, the objectives and knowledge gap are well articulated, the methods and results are clear, and the conclusions justified.
MORE SPECIFIC COMMENTS:
Comment, Table 5: Table 5 Mixed-effects models for frequency, duration, and magnitude of change water temperature metrics for all April to October 2016 data.
What does it add to the subject area compared with other published material? The sampling in analytical and the validity of questions is prominently in the paper. This statement should give a brief account of the purpose, need, and significance of the investigation. Improve it.
Response-We appreciate the suggestion and believe this comment refers to the Table 5 caption. We have added more detail to explain what the table shows and the caption now reads:
Table 5. Mixed-effects models to predict water temperature metrics based on April to October 2016 data for 14 study sites. The selected metrics are designed to quantify frequency and duration of exceedance of thermal stress thresholds and magnitude of thermal change for aquatic species in urban streams with 4 to 62% impervious land cover. The models listed in this table are the final, best fit model to predict each metric after testing 6 candidate models per metric. The best fit predictive models for frequency of exceedance of thermal stress temperatures include baseflow (BF) and % tree canopy in 5-meter riparian areas (TC5m) variables. The best fit predictive model for duration (Max days ≥ 20 °C) included BF, TC5m, and total length of stormwater pipes per loggershed (SWPipes). The best fit predictive models for magnitude of change (MaxT21, MaxT14, MaxT7) included TC5m and SWPipes.
Comment, Table 6: Table 6 Mixed-effects models for the monthly magnitude of change and variability water temperature metrics for the months of April to October 2016.
Decide how well the article presents the evidence that supports its argument. You must demonstrate that resilience to extreme temperature events. Do the results and discussion sections clearly summarize and interpret the data?
Response-The authors appreciate the suggestion and offer a more detailed caption to explain what the results in the table represent. The caption of Table 6 now reads:
Table 6. Mixed-effects models to predict water temperature metrics for magnitude of change and variability based on April to October 2016 data for 14 study sites. The average of daily maximum water temperature per month (Monthly ADMax), maximum range per month, and variance of the mean daily water temperature per month are designed to quantify the magnitude and variability in water temperature change per month for aquatic species in urban streams with 4 to 62% impervious land cover. The models listed in this table are the final, best fit model to predict each metric after testing 6 candidate models per metric. The predictive variables included in the best fit predictive models for monthly ADMax are variable by month, suggesting that sources of variability in urban stream habitats may affect aquatic species differently during times of spawning (April to May); growth of early life stages (June ); and times typically most thermally stressful (July to August). Final candidate models to predict water temperature variability were only significant for June variance and October range.
We have updated the Results and Discussion sections to better summarize and interpret the data.
Comment, Figure 3: You might observe that subjects in environ study didn’t accurately represent a diverse population. In Figure 3. you showed impervious and tree canopy land cover percentages within study site loggerheads (% Imp and % TC Lgshd), within 30-meter riparian areas (% Imp 30m 253 and % TC 30m), and within 5-meter riparian areas (% Imp 5m and % TC 5m).
Are these methods an appropriate, reasonable means of solving the problem? This point is only valid if researchers can point to recent developments in data gathering.
Response-We appreciate the reviewer’s feedback, we have updated Figure 3.
Comment, Figure 5: In Figure 5. you showed the monthly magnitude of change and variability per month for April to October.
The monthly mean ADMax, a maximum range of water temperature per month, and variance of mean daily water temperature per month and standard deviation values are shown. The results seem plausible in this part.
Methods are not detailed enough. Once described, they should evaluate the trends observed and explain the significance of the results to a wider understanding.
Response-The authors appreciate the reviewer’s feedback and have updated Figure 5. More detail has been added to the methods for Sections 2.4.2 and 2.4.3
Comment, Constructive feedback on Methods, Results, Conclusions:
Authors clearly outline the methodology and the expected result, but in the methodology section, you should more elucidate about. I have tried below to suggest some specific questions that, if answered, could illuminate both the message and the benefits of the paper.
Comment- The statistical and analytical methods are presented in a complete and clear format. However, small issues need further explanation.
Response-The authors agree with the suggestion to add further explanation to the methods and more detail has been added to the methods for Sections 2.4.2 and 2.4.3.
Comment-The conclusions and data interpretation are robust, valid and reliable. The authors have extended their conclusions beyond what their data actually shows, and the potential limitations of the study have been adequately discussed. However, the authors have no proposed relevant and reasonable recommendations based on their results. They should start by describing in simple terms what the data show
Response-The authors appreciate the acknowledgement that the data interpretation are robust and valid. We agree with the suggestion to better articulate in simple terms what the data actually show and have revised the Discussion and added a Conclusion section to the paper.
Comment-The study has helped to advance knowledge, but the novel results have not been appropriately highlighted. You should focus on some innovation in this manuscript. Please try to find literature elaborating river valley and ecological connection. This statement should give a brief account of the purpose, need, and significance of the investigation for instance ecosystem service. Improve it.
Response-The authors appreciate the acknowledgement that the study has helped to advance knowledge. We agree with the suggestion to better highlight novel results and innovation and have revised the Discussion and Conclusion to better highlight the significance of the investigation.
Comment-The precision of process data and regularity of sampling in time-dependent studies is likely to be a major flaw. Please give more details about the impact of urban patterns on aquatic ecosystems. Are enough examples provided to assist readers in relating to the author?
Response-The authors agree that the study is based on one year of data and that the “precision of process data and regularity of sampling in time-dependent studies is likely to be a major flaw” and considered when interpreting results from the study. We have added more discussion from the literature related to urban patterns and effects on aquatic ecosystems to help interpret and discuss results……as suggested.
Comment- You mentioned “Further research needed.- The most commonly applied thermal cooling best management practice (BMP) is riparian tree planting, a strategy that has been applied in the United States since the 1970s to mitigate the impacts from logging and agriculture [46,82– 84].
What are its implications for an understanding of scientific results by the general, literate public? In studies carried out over time are there sufficient data points to support the trends described by the authors?
Response-The authors added further details and revised the Discussion and Conclusion to address “implications for understanding scientific results”.
Comment- You reported that: “Our results were partially consistent with a previous study by Roy et al. (2005) that quantified tree canopy cover in riparian areas within 30-m widths along urban streams (≥ 15% impervious) that found no significant relationships to water temperature. The Roy et al. (2005) study was based on data collected in the field with riparian area 30-meter widths along 200-m reaches, whereas our study used geospatially extracted 5-meter and 30-meter riparian area data at the loggershed scale”.
You might add a chapter that describes a thermal habitat that restricts patterns of occurrence in multiple life-stages of a headwater for trout, dace, and crayfish species. How well does the modelling time-dependent effects of thermal stress on ectotherm growth rates summarize the article? The problem addresses, its techniques, results, and significance? Please specify. Remember that all of your ideas must support your central design.
Response-The authors have revised the Discussion and Lines 499-515 include content to address this comment.
Comment- You might examine that qualitative research, with sufficient descriptive elements and appropriate quotes from interviews unless strong arguments for such repetition are made. Add the impact of buffer zone or morphology or geomorphology of the riverbank in a river valley in an urban area. How the article relates to other work on the same subject? Are they supported by evidence and analysis for example River Habitat assessment in an urbanised river?
Sufficient use of control experiments in detailed methodology and the data analysis is done systematically (in qualitative research) it's all about the information gathered. In this paragraph, you might identify the question, how well the article resolves its central problem.
Response-We appreciate the suggestion and see why adding a discussion of literature from qualitative research and information from river habitat assessments may be valuable. However, we feel that this is out of the scope of the purpose and objectives for this paper and suggest this may be a valuable related study in the future.

Reviewer 2 Report
Dear Authors,
Please see the section-wise comments below on your research article with the following details.
Manuscript title: Riparian land cover, water temperature variability, and thermal stress for aquatic species in urban streams
Manuscript Number: water-1352243
Journal Submitted: Water
Specific Comments:
Title:
I guess the title can be revised somewhat like this “Riparian land cover can help relieve the water temperature mediated thermal stress in urban streams”
Abstract:
Please consider mentioning the effect of impervious surfaces on stream water temperature as well. The logical conclusions should be extended further to establish the main outcomes of this study.
Introduction:
This is an excellent part that successfully highlights the widening knowledge gap as well as delineates the necessity of such studies. I would suggest the authors summarize the leading impacts of varying temperature regimes in streams and particularly in the urban streams as these are becoming one of the most stressed ecosystems across the globe.
The research questions and objectives are clearly outlined, while the research question is justified given what is already known about the topic.
Materials and Methods:
The opening sentence must be revised.
The total number of sites appears to be 25 whereas the number of sites shown in figure 1, figure 2, and table 1 is far less than that. Please check this.
From the section of Correlation analysis, please create a separate sub-section as “Data analysis” and discuss all that under that section.
The study design is appropriate to answer the main aims of this study.
However, the authors need to mention enough details in order to replicate the study as well as if the method is reliable and valid.
Results:
You need to extend the description of figure 3 in the results section.
Figure 1. Please make sure the study sites are 25 as you mentioned at the start of the methods.
Figure 3. This is the most confusing figure. Please use distinct colors and different patterns the difference between the variables. This will help you and the readers to better understand the outcomes of this study.
Figure 5. The figure captions are usually placed at the bottom. This figure quality is low, hence should be improved.
The description of the mixed-effects models is too little to be sufficient. While this is the main concept of this study, therefore, you need to add more details.
Discussion:
Seeing the type of data presented and the overall nature of this study, I am convinced that the results and discussions should be combined as one section.
L 519-522: This appears to be a repetition of what was earlier described in the introduction section.
The results are discussed from multiple angles and placed into context without being overinterpreted. However, the authors should consider giving some aspects on the spawning and migration of the main targeted species as well.
Conclusions
There are no conclusions provided. Please create a new section and add the section “further research needed” in the same section as well.
References:
The main background reference Roy et al., 2005 is not cited and appears to be skipped in the reference citations.
Table 2. The references cited are not cited in the journal format. Please check and I guess you need to update your whole list of references.
Table 3. Same as above.
Please check other places in the whole MS for such errors e.g. L 176.
Author Response
Water-1352243 R1, Major Revisions Response
Title: Riparian land cover, water temperature variability, and thermal stress for aquatic species in urban streams
Submitted by:
Dr. Anne Timm, USDA Forest Service, Northern Research Station, Baltimore, Maryland, United States
Dr. Valerie Ouellet and Dr. Melinda Daniels, Stroud Water Research Center, Avondale, Pennsylvania, United States
Response to Reviewers
The authors greatly appreciate the feedback from the reviewers. We have carefully considered each comment and changed the manuscript accordingly. Below, we have responded to general comments and then individual, line by line comments from each reviewer. We have indicated what changes have been made in the manuscript for each comment. The latest version of the manuscript with edits has been uploaded.
MORE SPECIFIC COMMENTS:
TITLE
Comment- I guess the title can be revised somewhat like this “Riparian land cover can help relieve the water temperature mediated thermal stress in urban streams”
Current Manuscript title: Riparian land cover, water temperature variability, and thermal stress for aquatic species in urban streams
Response-The authors respectfully disagree with the title change because it explores how different types of riparian land cover can both increase and decrease water temperatures and variability
ABSTRACT
Comment- Abstract: Please consider mentioning the effect of impervious surfaces on stream water temperature as well. The logical conclusions should be extended further to establish the main outcomes of this study.
Response-The authors agree with the reviewer and have updated the abstract as follows:
Thermal regime warming and increased variability can result in human developed watersheds due to runoff over impervious surfaces and influence of stormwater pipes. This study quantified relationships between tree canopy, impervious surface, and water temperature in stream sites with 4 to 62% impervious land cover in their “loggersheds” to predict water temperature metrics relevant to aquatic species thermal stress thresholds. This study identified significant (≥0.7, P <0.05) negative correlations between water temperature and percent tree canopy in the 5-meter riparian area and positive correlations between water temperature and total length of stormwater pipe in the loggershed. Mixed-effects models predicted that tree canopy cover in the 5-meter riparian area would reduce water temperatures 0.01 to 6 °C and total length of stormwater pipes in the loggershed would increase water temperatures 0.01 to 2.6 °C. To our knowledge, this is the first time that the relationship between stormwater pipes and water temperature metrics has been explored to better understand thermal dynamics in urban watersheds. The results highlight important aspects of thermal habitat quality and water temperature variability for aquatic species living in urban streams based on thermal thresholds relevant to species metabolism, growth, and life history.
INTRODUCTION
Comment- This is an excellent part that successfully highlights the widening knowledge gap as well as delineates the necessity of such studies. I would suggest the authors summarize the leading impacts of varying temperature regimes in streams and particularly in the urban streams as these are becoming one of the most stressed ecosystems across the globe.
The research questions and objectives are clearly outlined, while the research question is justified given what is already known about the topic.
Response-The authors appreciate the acknowledgement that the introduction describes the knowledge gap and need for the study well and outlines and justifies research questions and objectives clearly.
The authors have included material on varying temperature regimes in streams on lines 36-42 and for urban streams on lines 43-52.
MATERIALS AND METHODS:
Comment- The opening sentence must be revised. The total number of sites appears to be 25 whereas the number of sites shown in figure 1, figure 2, and table 1 is far less than that. Please check this.
Response-The authors thank the reviewer for pointing out this error and have replaced 25 with 14 on line 87 and on line 149, which is the correct number that is also consistent with Figures 1 and 2 and Table 1.
Comment- From the section of Correlation analysis, please create a separate sub-section as “Data analysis” and discuss all that under that section.
Response-The authors agree with this suggestion and have added subsection headings throughout the methods section, including adding the Data analysis subsection, 2.4.
Comment- The study design is appropriate to answer the main aims of this study.
However, the authors need to mention enough details in order to replicate the study as well as if the method is reliable and valid.
Response-The authors agree and have added more details to the methods, sections 2.4.2 and 2.4.3.
RESULTS AND FIGURES:
Comment- You need to extend the description of Figure 3 in the results section.
Response-The details on what is presented in Figure 3 can be found in the Results section, Lines 250-256.
Comment- The description of the mixed-effects models is too little to be sufficient. While this is the main concept of this study, therefore, you need to add more details.
Response-The authors agree and have updated Sections 2.4.2 and 2.4.3.
Comment- Figure 1. Please make sure the study sites are 25 as you mentioned at the start of the methods.
Response-The authors have changed the number of study sites to 14 and have reformatted sentences to reflect the correct numbers on Lines 87 and 149.
Comment- Figure 3. This is the most confusing figure. Please use distinct colors and different patterns the difference between the variables. This will help you and the readers to better understand the outcomes of this study.
Response-The authors agree that this figure is hard to read and have increased the font and changed it to a top and bottom panel style. The top panel of the figure includes tree canopy land cover and the bottom panel of the figure includes impervious land cover.
The caption for Figure 3 now reads as follows:
Figure 3. Tree canopy land cover percentages (top panel) and impervious land cover percentages (bottom panel) within study site loggersheds (% TC Lgshd and % Imp Lgshd), within 30-meter riparian areas (% TC 30m and % Imp 30m), and within 5-meter riparian areas (% TC 5m and % Imp 5m).
Comment- Figure 5. The figure captions are usually placed at the bottom. This figure quality is low, hence should be improved.
Response-The authors appreciate the reviewer bringing this to our attention and have moved all figure captions throughout the paper to the bottom. The authors have also redone Figure 5 as a higher resolution version as suggested.
DISCUSSION:
Comment- Seeing the type of data presented and the overall nature of this study, I am convinced that the results and discussions should be combined as one section.
Response-The authors have added more interpretation to the Results and have moved material that was in the Discussion to the Results.
Comment- L 519-522: This appears to be a repetition of what was earlier described in the introduction section.
Response-The authors agree that these sentences are repetitive material from the introduction and have been deleted.
Comment- The results are discussed from multiple angles and placed into context without being overinterpreted. However, the authors should consider giving some aspects on the spawning and migration of the main targeted species as well.
Response-The authors included material on spawning, lines 501-517.
CONCLUSIONS:
Comment- There are no conclusions provided. Please create a new section and add the section “further research needed” in the same section as well.
Response-The authors have added a conclusion section, now Section 5, as suggested and have moved the “further research needed” section to the conclusion section.
REFERENCES:
Comment- The main background reference Roy et al., 2005 is not cited and appears to be skipped in the reference citations.
Response- We thank the reviewer for bringing this to our attention. We have updated the Roy et al. 2005 reference as the correct format.
Comment- Table 2. The references cited are not cited in the journal format. Please check and I guess you need to update your whole list of references.
Response-We thank the reviewer for bringing this to our attention. We have updated the reference formats in Table 2 and updated errors in other references throughout the manuscript.
Comment- Table 3. Same as above. The references cited are not cited in the journal format. Please check and I guess you need to update your whole list of references.
Response-We thank the reviewer for bringing this to our attention. We have updated the reference formats in Table 3.
Comment- Please check other places in the whole MS for such errors e.g. L 176.
Response- We thank the reviewer for bringing this to our attention. We have updated the reference formats throughout the manuscript.

Round 2
Reviewer 1 Report
Thanks for correcting most of the mistakes. The text has become revised and improved.
Reviewer 2 Report
Excellent. Job well done.